# A Novel Cationic Polymer Surfactant for Regulation of the Rheological and Biocidal Properties of the Water-Based Drilling Muds

**DOI:** 10.3390/polym15020330

**Published:** 2023-01-09

**Authors:** Kaldibek Abdiyev, Milan Marić, Baurzhan Orynbaev, Mariamkul Zhursumbaeva, Nurgul Seitkaliyeva, Zhexenbek Toktarbay

**Affiliations:** 1Department of Chemical Processes and Industrial Ecology, Satbayev University, 22 Satbayev Str., Almaty 050013, Kazakhstan; 2Department of Chemical Engineering, McGill University, 3610 University Str., Montreal, QC H3A 0C5, Canada; 3Renewable Energy Systems and Material Science Laboratory, National Laboratory Astana (NLA), Nazarbayev University, 53 Kabanbay Batyr, Astana 010000, Kazakhstan

**Keywords:** copolymer, structure-formation, biocidal polymers, antibacterial activity, sulfate-reducing bacteria, rheology

## Abstract

The copolymer of N,N-diallyl-N,N-dimethylammonium chloride and N-[3-(Dimethylamino)propyl]methacrylamide (DADMAC–DMAPMA) was synthesized by radical polymerization reaction in an aqueous solution in the presence of the initiator ammonium persulfate (NH_4_)_2_S_2_O_8_. The molar compositions of the synthesized copolymers were determined using FTIR and ^1^H NMR-spectroscopy, elemental analysis, and conductometric titration. It was found that in the radical copolymerization reaction, the DMAPMA monomer was more active than the DADMAC monomer; for this reason, the resulting copolymers were always enriched in the DMAPMA monomers. The study of the influence of the DADMAC–DMAPMA copolymer on structure-formation in the bentonite suspension showed that this copolymer significantly increased the static shear stress (SSS) of the system. In this case, the structure-forming properties of the copolymer depended on the pH of the medium. The lower the pH level, the better the structure-formation was in the suspension in the presence of the copolymer. The study of antibacterial activity showed that the DADMAC–DMAPMA copolymer had a biocidal effect against sulfate-reducing bacteria (CRB) at a concentration of not less than 0.05 wt.% and can be used to inhibit the growth of this bacteria.

## 1. Introduction

In the oil and gas industry, the successful drilling of wells largely depends on the correct choice of drilling fluid [1,2]. The drilling fluid must perform, first of all, the following functions: (1) remove cuttings from under the bit; (2) transport cuttings up to the surface; (3) keep cuttings particles in suspension while stopping the circulation of the solution; (4) cool the bit and facilitate rock destruction in the bottom hole zone; (5) create pressure on the walls of the well to prevent water, oil and gas shows; (6) provide a physico-chemical effect on the well wall to prevent its collapse; (7) transfer energy to the hydraulic downhole motor (when drilling with these motors); and (8) ensure the preservation of the permeability of the producing formation during its opening [3,4,5,6].

Drilling fluids are divided into the following types: water-based muds (WBMs), oil-based muds (OBMs) and foam drilling fluids [1,5,6,7]. Among them, water-based drilling fluids are widely used. This is due to the fact that water-based drilling fluids are relatively cheap and environmentally safer [1,5,6,7,8,9,10]. However, water-based drilling fluids have several disadvantages, one of which is the erosion of the well wall and the absorption of water by the formation, which may result in the collapse of the well wall and the swelling of the rock.

To perform the above basic functions, drilling fluids must have some necessary rheological properties (plastic viscosity, yield strength, thixotropy and gel strength), stability to temperature and pressure changes, resistance to the action of contaminating components (drilled rock, salt reservoir water, electrolytes), and they should prevent fluid loss additives [2,11,12]. In order to provide the necessary rheological properties, various substances (bentonite clays, water-soluble polymers, surfactants, electrolytes and other substances) are added to the water-based drilling fluids [1,9,12,13,14].

Bentonite clay, added to the drilling fluid, provides the necessary viscosity in the system, volume structure formation, thixotropic properties of the suspension, and the formation of a thin filtration layer on the surface of the well wall. A filtration layer reduces the risk of collapse of the well wall and decreases water filtration through the well walls into the formation (reservoir water absorption).

The filtration characteristics of a water-based drilling fluid have a significant impact on wellbore stability. The penetration of water into porous formations reduces the stability of the wellbore and can lead to serious problems, such as a stuck pipe and the collapse of formations [6,13,14].

The viscosity and thixotropic nature of the drilling fluid ensure the retention of cuttings particles in the suspension during a temporary (forced) stop of the drilling process, i.e., prevent sedimentation of drill cuttings particles on the bit of the drilling rig.

In many cases, the addition of natural and synthetic water-soluble polymers to the clay drilling fluids ensures the formation of a thin filter cake on the wellbore wall, which sharply reduces water filtration through the wellbore walls into the formation and increases the stability (strength) of the wellbore wall [1,12,14,15,16,17,18,19].

To improve the rheological and filtration properties of the drilling fluids, the following natural polymeric compounds were used: carboxymethyl cellulose and polyanionic cellulose [20,21,22], xanthan gum [23,24,25], wild Jujube pit powder [26], tea polyphenols [27], starch and its derivatives [28,29,30], bread fruit starch, cassava starch, bush mango seed, corn fiber [31], and pomelo peel powder [32].

The main advantage of natural polymer compounds is their availability and relatively low cost. However, natural polymeric additives to drilling fluids based on polysaccharides have a serious drawback—their high biodegradability under the action of various microorganisms [33,34]. As a result of the vital activity of cellulose decomposing and other types of bacteria, the rheological and filtration characteristics of drilling fluids prepared on the basis of polysaccharides can significantly deteriorate [34].

To suppress the vital activity of microorganisms, different substances with a bactericidal effect, such as phenols, formaldehyde, dialdehyde (glyoxal), ketones [33], and polymeric bactericides [35,36,37], are added to the drilling fluids.

Another serious problem in the oil and gas industry is microbiological corrosion of drilling rigs, and oil and gas constructions [38,39,40]. Due to biological corrosion, several thousand tons of metal products and structures fail every year. Usually, metal constructions are not affected by microorganisms. However, during vital activity (metabolism) of microorganisms, some corrosive-dangerous substances (hydrogen sulfide, acids, bases) can be released into the environment. Among bacteria, the most dangerous in terms of corrosion are sulfate-reducing bacteria (SRB) (Desulfovibrio and Desulfotomaculum species) and thionic (TB) (Tiobacilus species) [40,41,42].

Biocidal polymers are often used to inhibit chlorine-resistant bacteria (CRB) [43,44,45]. However, these compounds are not always effective. According to [46,47], regular inhibition of the CRB with the same compounds can promote the development of biocide-resistant mutants. For this reason, mixtures of biocides are often used to prevent the development of mutant strains. Today, despite the many biocidal compounds, microbiological corrosion of metal constructions and equipment remains one of the unsolved problems.

In this regard, the synthesis of new water-soluble polymer surfactants from available industrial monomers, which simultaneously have structure-forming and biocidal properties, is one of the urgent tasks.

In this work, a new polymeric surfactant has been synthesized, which has a structure-forming and biocidal effect.

## 2. Experimental

### 2.1. Materials

The solution of the monomer N,N-diallyl-N,N-dimethylammonium chloride (DADMAC) (65 wt% in water, d = 1.056 g/mL), monomer N-[3-(Dimethylamino)propyl]methacrylamide (DMAPMA) (purity 99%, d = 0.94 g/mL at 25 °C) and 1,4-Dioxane (purity > 99%, d = 1.034 g/mL) were purchased from Sigma Aldrich (USA).

AgNO_3_ (purity 99.8 wt%) and ammonium persulfate, (NH_4_)_2_S_2_O_8_ (purity 99.7 wt%) were kindly provided by “LaborPharma” Ltd. (Almaty, Kazakhstan). Acetone (purity 99.9 wt%) was purchased from “Labchimprom” Ltd. (Almaty, Kazakhstan). Argon (purity 99.995 wt%) was purchased from “Ikhsan TechnoGas” Ltd. (Almaty, Kazakhstan). Distilled water with a conductivity of 2.4 mS/cm (20 °C) was used in all experiments.

E from the Tagan deposit (TAGBENT Ltd, Oskemen city, Kazakhstan) was used to study the structure-formation properties of the copolymers. Chemical analysis of the used bentonite sample was carried out by X-ray fluorescence spectroscopy (XRF) using PANalytical Axios PW400. Bentonite clay had the following composition, wt.%: Na_2_O-1.00; MgO-3.59; Al_2_O_3_-22.46; SiO_2_-58.78; SO_3_-0.06; Cl-0.02; K_2_O-0.59; CaO-3.99; TiO_2_-0.79; Cr_2_O_3_-0.01; Fe_2_O_3_ 8.60 and others–0.2 [48].

The specific surface area of the bentonite clay was determined in liquid N_2_ by the Brunauer–Emmett–Teller (BET) approach using a Micromeritics ASAP 2020 instrument at −196 °C (77 K). The approximate total pore volume of the clay was calculated from the amount of adsorbed liquid at 0.99 relative pressures. Distribution of pore sizes was calculated from the desorption part of the N_2_ adsorption/desorption isotherms, using the Barrett–Joyner–Halenda (BJH) method [48].

Bentonite clay particles had the following characteristics: BET surface area–59 m^2^/g; Langmuir surface area–74 m^2^/g; t-Plot micropore area–2 m^2^/g; t-Plot external surface area–57 m^2^/g; BJH desorption cumulative surface area of pores with diameter 2–100 nm–57 m^2^/g; Single point adsorption total volume of pores with diameter less than 141 nm at P/P_0_ = 0.99 − 0.39 cm^3^/g; t-Plot micropore volume–0.0003 cm^3^/g; BJH Desorption cumulative volume of pores with diameter 2–100 nm–0.38 cm^3^/g; Adsorption average pore width (4 V/A) by BET–26 nm and BJH desorption average pore diameter (4 V/A)–27 nm [48].

Before use, the bentonite clay was ground in a ball mill at 300 rpm for 15 min and then sieved through a sieve with a pore diameter of 0.01 mm. For the study, a sample with a particle diameter of not more than this value was used.

### 2.2. Methods

#### 2.2.1. Spectrometric Methods

FTIR spectra of the copolymer sample was obtained on an Avatar 370 CsJ FTIR spectrometer in the spectral range of 4000–400 cm^−1^ from tablets prepared by compressing a 2 mg sample with 200 mg of KBr. The ^1^H NMR spectra was recorded on a JNN-ECA Jeol 400 spectrometer (JEOL Ltd, Tokyo, Japan) (frequency 400 MHz) using D_2_O as a solvent at 20 °C. Chemical shifts were measured relative to signals of residual protons of the deuterated solvent.

Elemental analysis of the copolymer samples for H, C and N was performed on a Vario El cube instrument (Elementar Ltd, Langenselbold, Germany). Conductometric titration of aqueous solutions of the copolymers was carried out at room temperature with an AgNO_3_ solution and by using the Metrohm 856 Conductivity Module.

#### 2.2.2. Determination of the Monomer Reactivity Ratios

The reactivity ratio is an important parameter for describing the copolymerization mechanism and composition of copolymers. The monomer reactivity ratio can be determined from the initial monomer ratio and resulting copolymer composition by the Mayo–Lewis (1) and Fineman–-Ross Equations (2) [49,50]:r_2_ = F⋅{(1/f)⋅(F⋅r_1_ + 1) − 1} (1)
(F/f)⋅(f − 1) = r_1_⋅(F^2^/f) − r_2_ or y = r_1_⋅x − r_2_
(2)
where F = [M_1_]/[M_2_]; f = [m_1_]/[m_2_]; [M_1_] and [M_2_] are mole fractions of DMAPMA and DADMAC in the initial monomers’ mixture, respectively; [m_1_] and [m_2_] are mole fractions of DMAPMA and DADMAC in the resulted copolymer, respectively; y = (F/f)⋅(f − 1); x = F^2/^f; r_1_ and r_2_ are reactivity ratios of DMAPMA and DADMAC, respectively.

#### 2.2.3. Structure-Formation Test

Structure formation in the bentonite suspensions in the presence of a copolymer was determined using a rotational viscometer with manual control of 6 speeds RC-35D (RIGCHINA com, Zhejiang, China). For the study, 30 wt.% suspension of the bentonite clay was prepared by stirring at 300 rpm for 10 min in water. Then, the prepared suspension was left for 24 h for the complete swelling of the bentonite clay. The next day, the suspension was treated with the required volume of copolymer solution and stirred at 100 rpm for 3 min. After treating the suspension with the copolymer, the concentration of the suspension was always equal to 20 wt.%. A copolymer solution with a concentration of 0.1 wt.% was prepared the day before. The desired copolymer concentration was prepared by diluting the stock solution.

The static shear stress (SSS) of the bentonite suspensions was measured at room temperature, 1 min (SSS-1) and 10 min (SSS-10) after the treating of the suspension with the copolymer solution. The SSS is one of the most important drilling fluid parameters. It indicates the rate of the formation and strength of the gel forming in suspension at rest. The higher the SSS, the larger the size of the particles that are in suspension and that do not settle to the bottom of the well.

#### 2.2.4. Antimicrobial Action of the DADMAC–DMAPMA Copolymer against SRB Investigation

The antimicrobial action of the DADMAC–DMAPMA copolymer, with different molar compositions, on sulfate-reducing bacteria (SRB) was tested at the “Scientific and Production Center of Microbiology and Virology, LLP” (Almaty, Kazakhstan). The effect of the copolymer samples on the growth of SRB was studied in three concentrations (wt%): 0.01, 0.05 and 0.1.

For the study, 1 mL of the culture liquid of SRB was added to the test tubes. Then, the tubes were filled with a nutrient medium with a copolymer dissolved in the appropriate concentration. In the control sample, SRB was cultivated without copolymer. When SRB grows, a black film with a metallic outflow on the inner surface of the tube is observed.

#### 2.2.5. Synthesis

The copolymer of N,N-diallyl-N,N-dimethylammonium chloride and 3-[(dimethyl)aminopropyl]methacrylamide (DADMAC–DMAPMA copolymer) was synthesized in the ampoule by radical copolymerization of DADMAC with DMAPMA in an aqueous environment at different initial molar ratios of monomers in the presence of (NH_4_)_2_S_2_O_8_ as an initiator. The total molar concentration of monomers in the mixture was equal to 5 mol/L. After purging with argon for 20 min, the ampoule containing the monomers was sealed and placed in a water bath heated at 343 K. Then, the reaction mixture was heated for 180 min. After this, the resulting copolymer was precipitated from the reaction environment and repeatedly purified with a mixture of 1,4-dioxane and acetone (50:50 vol.%). At the end, the purified copolymer was dried under a vacuum at 313 K until a constant weight was achieved. The percentage conversion (y) of the monomers was obtained by gravimetry (the mass ratio of the isolated copolymer to the initial monomers’ mass).

Determination of the copolymerization reaction rate (*R_c_*).

The value of *dy*/*dt* was obtained from the slop of the plot of *y* versus *t*. *R_c_* was calculated according to Equation (3) [51]:(3)$$R_{C}=-\frac{d\left[M\right]}{dt}=\left[M\right]\frac{dy}{dt}$$
where [M] is the total monomers’ concentration; *t* is the time.

## 3. Results and Discussion

### 3.1. Determination of Copolymer Composition

According to previous studies [52,53,54], in the radical polymerization reaction, the DADMAC monomers react with both bonds to form a five-membered cyclic structure in a macrochain, because the elementary act of this structure formation is sterically more favorable than that of a six-membered cyclic structure formation. At the same time, considering side reactions [55,56], we cannot exclude the possibility of a copolymerization reaction occurring at one double bond of the DADMAC monomer and, accordingly, the presence of an unreacted double bond CH_2_ = CH− in the structure of the obtained copolymer. According to [57], poly-DADMAC synthesized by radical polymerization in water contains up to 3% unreacted double bonds. Therefore, we can propose that the DADMAC–DMAPMA copolymer can have the following two different molecular structures (Figure 1):

#### 3.1.1. The Results of Infrared Spectroscopic Studies

The FTIR spectrum of the DADMAC–DMAPMA copolymer (Figure 1) synthesized from an equimolar monomer mixture contains a wide band at 3376 cm^−1^, characteristic of the NH-group of the secondary amide DMAPMA, as well as the quaternary ammonium group DADMAC and the tertiary amino group DMAPMA. In the area of 1650 cm^−1^, the absorption band characteristic of the carbonyl group C=O of the DMAPMA monomer is detected. The peaks at 2974 cm^−1^ and 1550 cm^−1^ correspond to the stretching and bending vibrations of the methyl group. The absorption bands in the area of 1500–500 cm^−1^ can be attributed to the stretching vibrations of simple C–C bonds and to the deformation vibrations of simple C–H and N–H bonds [58].

#### 3.1.2. Analysis of the NMR Spectrum of the DADMAC–DMAPMA Copolymer

To identify the composition of the DADMAC–DMAPMA copolymer obtained from an equimolar mixture of monomers, ^1^H NMR spectra of the homopolymers poly-DMAPMA (Figure 2) and poly-DADMAC (Figure 3) were chosen. To determine the molar composition of the DADMAC–DMAPMA copolymer (Figure 4), absorption bands that belong to the structural units of the DMAPMA and DADMAC monomers were chosen. In the spectrum of the DADMAC–DMAPMA copolymer (Figure 4) synthesized from the monomers’ mixture with molar composition [DMAPMA]:[DADMAC] = 65:35, the absorption band at 2.09 ppm was selected for the content of the functional structural units of DMAPMA (Figure 2), which is responsible for the content of protons H-1 in the copolymer.

In the homopolymer-poly-DMAPMA, this proton appears at 2.03 ppm. (Figure 2). In the spectrum of the DADMAC–DMAPMA copolymer (Figure 4), the content of the functional structural units DADMAC is responsible for, appear in the absorption band at 3.72–3.74, protons H-10, 10^1^.

In the homopolymer poly-DADMAC, these protons appear at 3.69–3.71 ppm (Figure 3). The integral proton intensity of protons H-1 in the copolymer is 45.71, and the integral proton intensity of protons H-10, 10^1^ in the copolymer equals 28.73 (Figure 4). The molar ratio of the structural units is as follows: DMAPMA, mol. = 45.71:2 = 28.86; DADMAC, mol. = 28.73: 4 = 7.18. Then, DMAPMA:DADMAC = 28.86:7.18 = 4.0:1.0.

So, the molar composition of the copolymer synthesized from the mixture DMAPMA, 65 mol % mol + DADMAC, 35 mol % is equal to 80:20. This value is very close to the composition determined by elemental analysis and conductometric titration (76:24).

### 3.2. Monomer Reactivity Ratios

The reactivity ratios of the monomers DMAPMA and DADMAC were determined using the Mayo–Lewis (1) and Fineman–Ross (2) equations for the conversion of not more than 10% [49,50] (Table 1 and Table 2).

The graphical plots of the reactivity ratio determined by the Mayo–Lewis and Fineman–Ross methods are given in Figure 5 and the data are summarized in Table 3.

A comparison of the reactivity ratios of the monomers showed that the DMAPMA monomer is more active than DADMAC in the radical copolymerization reaction. Furthermore, the value of r_1_ is more than 1, while the value of r_2_ is less than 1 in each performed case. This means that for all concentration ratios of the monomers in the reaction mixture, the resulting copolymer is enriched in the DMAPMA monomer.

### 3.3. Results of Elemental Analysis and Conductometric Titration

The results of the elemental analysis of the copolymer samples for H, C and N and the conductometric titration of aqueous solutions of the DADMAC–DMAPMA copolymer with AgNO_3_ are presented in Table 4.

As can be seen, the results of the elemental analysis and conductometric titration practically match each other. It should be noted that in the radical copolymerization reaction in the mixture of DADMAC with DMAPMA, the DMAPMA monomer is more active than DADMAC; for this reason, the resulting copolymer is always enriched in the DMAPMA monomer.

The relatively high activity of the DMAPMA monomer in the radical copolymerization reaction can be explained by the stability of the macroradical terminal DMAPMA unit, because of the resonance with a carbonyl group (Figure 2) [54]:

The relatively low reactivity of the DADMAC monomer can be explained by the complex structure of this monomer and the presence of a positive charge in the macrochain. Like-charged monomers repel each other when they approaching, which makes it difficult for them to interact with each other closely enough to react. The relatively low reactivity of the DADMAC monomer in radical copolymerization reactions with dimethylacrylamide, acrylonitrile, itaconic and fumaric acids, and 4-vinylpyridine was noted in several works [53,54,59,60,61,62,63,64].

### 3.4. Study of the Influence of the Initiator and Monomers’ Concentration on the Kinetics and the Conversion of the Copolymerization Reaction of DADMAC with DMAPMA

Figure 6 shows the conversion dependence on the copolymerization reaction’s duration at different initiator concentrations.

As shown in Figure 6, with an increase in the concentration of the initiator, the rate and conversion of the copolymerization reaction increase. From the figure, the rate (R_c_) and conversion (y, %) (after 240 min) of the reaction were determined (Table 5). The rate of the copolymerization reaction was calculated according to Equation (1).

As can be seen from Table 5, with an increase in the concentration of the initiator from 0.05 to 1.0 wt%, the rate and conversion of the copolymerization reaction increases by approximately two times. This phenomenon can be explained as follows: an increase in the concentration of the initiator leads to an increase in the number of free radicals in the reaction medium, as they are the active centers of the copolymerization reaction.

From the log *R_c_* = n log [I] plot (Figure 7), the equation for the dependence of the reaction rate on the initiator concentration was determined, which looked as follows: *R_c_* ∝ [I]^0.033^.

Figure 8 shows the influence of the total monomer concentration on the percentage conversion of the copolymerization reaction. As can be seen from the figure, with an increase in the total monomer concentration, the rate and conversion of the copolymerization reaction increase.

Then, from the figure, the rate (R_c_) and conversion (y, %) (after 240 min) of the reaction were determined (Table 6). The rate of the copolymerization reaction was calculated according to Equation (1).

As can be seen from Table 6, with an increase in the total monomer concentration from 1.5 to 6.0 mol/L, the rate of the copolymerization reaction increases by approximately 100 times, and the conversion of the reaction increases by approximately 15 times. This can be explained as follows: with an increase in the concentration of monomers in the reaction medium, the probability of the collision of the monomers with active free radicals increases. All this leads to an increase in the rate and conversion of the copolymerization reaction.

From the log *R_c_* = n log [M] plot (Figure 9), the equation for the dependence of the reaction rate on the total monomer concentration was determined, which looked as follows: *R_c_* ∝ [M]^3.48^.

Thus, we can conclude that the copolymerization rate equation for the DADMAC–DMAPMA copolymer is *R_c_* = *k* [M]^3.48^ × [I]^0.033^ (*k* is a specific rate, [M] and [I] are the monomer and initiator concentrations, respectively) for T = 343 K and the initial reaction mixture [DADMAC]:[DMAPMA] = 50:50 mol %.

Exponent numbers for a total monomer concentration greater than 1.0 have been found in many studies. In the paper by Liu and co-workers [51], the reaction equation for the copolymerization of DADMAC with acrylamide (AA) at 55 °C in the presence of (NH_4_)_2_S_2_O_8_ was obtained. The copolymerization rate equation for this system was R_p_ = *k* × [M]^2.23^ × [I]^0.70^, where [DADMAC]:[AA] was 4:1.

The polymerization rate equation for the copolymer DADMAC–VEMEA (Vinyl Ether of Monoethanolamine) at 65 °C (initiator-(NH_4_)_2_S_2_O_8_) was obtained in [35]: R_p_ = *k* × [M]^2.6^ × [I]^0.60^ (where [DADMAC]:[VEMEA] = 90:10 in initial reaction mixture).

Jaeger W, et al. [57], studied the kinetics of polymerization of DADMAC in the presence of (NH_4_)_2_S_2_O_8_) as an initiator and obtained the following overall rate equation: R_p_ = *k* × [M]^2.9^ × [I]^0.8^.

Zhang W, et al. [61], studied the kinetics of copolymerization of DADMAC and AA initiated by an Na_2_S_2_O_8_-Na_2_SO_3_ redox system at 40 °C in an aqueous solution and obtained the following polymerization rate equation: R_p_ = *k* × [Na_2_S_2_O_8_]^0.55^ × [Na_2_SO_3_]^0.56^ × [M]^1.14^ for m(DADMAC):m(AA) = 1:2 (the mass ratio of DADMAC to AA in the initial monomer solution).

In the study [51], the kinetics of the polymerization of DADMAC and AA with different monomer molar ratios initiated by an (NH_4_)_2_S_2_O_8_-Na_2_SO_3_ redox complex in an aqueous solution were studied. At 45 °C for the monomer molar ratio n(DADMAC):n(AA): 1:9; 2:8; 3:7; 4:6 and 5:5, the following copolymerization rate equations were obtained, respectively: R_p1_ = *k* × [M]^2.61^ × [I_O_]^0.51^ × [I_R_]^0.52^; R_p2_ = *k* × [M]^2.70^ × [I_O_]^0.50^ × [I_R_]^0.53^; R_p3_ = *k* × [M]^2.73^ × [I_O_]^0.50^ × [I_R_]^0.56^; R_p4_ = *k* × [M]^2.77^ × [I_O_]^0.51^ × [I_R_]^0.59^ and R_p5_ = *k* × [M]^2.84^ × [I_O_]^0.51^ × [I_R_]^0.61^ (where [M] is the total monomer concentration, and [I_O_] and [I_R_] are the oxidant and reductant concentrations, respectively) [59].

The authors of [62], studied the copolymerization reaction of DADMAC with dimethylacrylamide (DMAA) in an aqueous media in the presence of ammonium persulfate as an initiator at 60 °C and obtained the following copolymerization rate equation: R_p_ = *k* × [M]^2.63^ × [I]^0.40^.

According to [51,63], the exponent of [M] greater than 1.0 may have been due to the complicated interaction between comonomers, the ion-pair, and polarity effects.

Now, using the resulting equation: *R_c_* = *k* × [M]^3.48^ × [I]^0.033^, we can calculate the rate constant (*k*) of the copolymerization reaction for the system DADMAC–DMAPMA. From Table 4, at a fixed [I] = 0.5 wt% ≈ 0.022 mol × L^−1^, the value *R_c_* = 0.068 mol × L^−1^ × min^−1^ corresponds to the value [M] = 4 mol × L^−1^. Then we have 0.068 mol × L^−1^ × min^−1^ = *k* × [4 mol × L^−1^]^3.48^ × [0.022 mol × L^−1^]^0.033^; *k* = 1.03 × 10^−5^ L^2.51^ × mol^−2.51^ × s^−1^.

From Table 3, at a fixed [M] = 4 mol × L^−1^, the value *R_c_* = 0.067 mol × L^−1^ × min^−1^ corresponds to the value [I] = 0.5 wt% ≈ 0.022 mol × L^−1^. Then we have 0.067 mol × L^−1^ × min^−1^ = *k* × [4 mol × L^−1^]^3.48^ × [0.022 mol × L^−1^]^0.033^; *k* = 1.02 × 10^−5^ L^2.51^ × mol^−2.51^ × s^−1^.

Thus, the average value of the rate constant for the copolymerization reaction of DADMAC with DMAPMA at 70 °C in the presence of ammonium persulfate is *k* = 1.015 × 10^−5^ L^2.51^ × mol^−2.51^ × s^−1^.

In the paper [35], the kinetics of the copolymerization of DADMAC with VEMEA in an aqueous solution at 65 °C (initiator - (NH_4_)_2_S_2_O_8_) was studied and obtained the following value: *k* = (0.85÷2) × 10^−5^ L^2.2^ × mol^−2.2^ × s^−1^.

The authors of [60], studied the kinetics of the copolymerization of DADMAC with AA in an aqueous medium at 60 °C in the presence of H_2_O_2_ as an initiator. It was found that the rate of the copolymerization reaction depended on the temperature, and at 60 °C, it was equal to 4.8 × 10^−5^ mol × L^−1^ × s^−1^.

In other papers [64,65], the kinetics of the copolymerization of DADMAC with a vinyl ether of ethylene glycol at 70 °C in the presence of initiator (K_2_S_2_O_8_) was studied. It was found that by increasing the initiator concentration from 1.3 × 10^−2^ to 1.75 × 10^−2^ mol × L^−1^, the rate of the copolymerization reaction increased from 1.13 × 10^−4^ to 1.28 × 10^−4^ mol × L^−1^ × s^−1^.

### 3.5. Structure-Formation Properties of the DADMAC–DMAPMA Copolymers

In order to determine a possibility of using the DADMAC–DMAPMA copolymer as a structure-forming agent for controlling the viscosity (thixotropic properties) of clay drilling fluids, we studied the static shear stress after 1 (SSS-1) and 10 min (SSS-10) aging of a bentonite clay suspension of the Tagan deposit (West Kazakhstan) in the presence of poly-DADMAC and DADMAC–DMAPMA copolymers with different compositions (Figure 10, Figure 11 and Figure 12). Analysis of the figures showed that an increase in the concentration of poly-DADMAC leads to a slight increase in the SSS of the suspension; while with an increase in the content of DMAPMA in the DADMAC–DMAPMA copolymer up to 80 mol %, a sharp increase in the SSS of the suspension is observed. In addition, it should be noted that SSS depends on the pH of the medium. In an acidic medium, the structure-formation properties of the DADMAC–DMAPMA copolymer are enhanced (Figure 12).

According to [66,67], the particles of bentonite suspension are negatively charged and their zeta potential changes from −32 to −40 mV when the pH changes from 7 to 8 [63]; while the macromolecules of the DADMAC–DMAPMA copolymer have a positive charge due to the presence of DADMAC monomers in the copolymer composition (Figure 1). Thus, when the copolymer is added to the bentonite suspension, positively charged copolymer macromolecules are adsorbed onto the surface of the bentonite particles via an electrostatic mechanism [66,67]. The cooperative electrostatic interaction between copolymer chains and suspension particles occurs according to the polymer complex formation and bridging mechanism [66,67,68,69]. As a result, the surface charge of the bentonite particles decreases and the interaction between the bentonite suspension particles is enhanced. This ultimately leads to an increase in structure formation in the bentonite suspensions and an increase in SSS.

An increase in the content of DMAPMA in the copolymer raises the hydrophobicity of the macromolecule, which leads to enhanced stability of the copolymer–bentonite particles’ association [69,70].

The increase in the suspension’s SSS in an acidic medium can be explained as follows. In the acidic medium, a protonization of the amino groups of the DMAPMA monomers takes place, and as a result, the positive charge density (zeta potential) of the DADMAC–DMAPMA copolymer significantly increases. Due to this, the electrostatic interaction of the copolymer chains with the surfaces of the bentonite particles is enhanced.

### 3.6. Investigations of the Antibacterial Activity of the DADMAC–DMAPMA Copolymer against the Growth of SRB

The DADMAC–DMAPMA copolymers synthesized from mixtures of monomers with the molar composition [DMAPMA]:[DADMAC] = 80:20 (A), 50:50 (B), and 20:80 (C), were chosen as biocidal materials.

The effect of copolymer samples on SRB growth was studied at three concentrations: 0.1% (1), 0.05% (2), and 0.01% (3). The development of SRB was observed on the fifth day of cultivation. When SRB grows, a black film with a metallic outflow on the inner surface of the tube appears. It was found that the samples of DADMAC–DMAPMA copolymer (1) and (3) completely (100%) inhibited the development of SRB at concentrations of 0.05% and 0.1%; however, at a concentration of 0.01%, the development of SRB was observed. Sample (2) completely inhibited the development of SRB, only at a concentration of 0.1%. The biocidal effect of the samples was observed throughout the test period (for 15 days).

According to [71,72,73,74], the surfaces of most microbial cells and bacteria have a total negative charge. At the same time, many polycations and polyampholytes with antimicrobial properties are known [48,75,76,77,78,79,80]. Therefore, considering this, we can propose the following mechanism for suppressing the growth of SRB in the presence of a polycation-DADMAC–DMAPMA copolymer. When a copolymer is added to a medium with SRB, the macromolecules of the copolymer are adsorbed on the bacterial surface by an electrostatic mechanism. As a result, a thick adsorption layer, with a certain thickness, strength, elasticity, and stabilized by hydrophobic interactions between the hydrocarbon radicals of the polymer, can form on the surface of the bacterial membrane. The adsorption layer on the surface of microorganisms limits the access of oxygen and nutrients to the interior of the microorganism. All this ultimately leads to the death of the microorganism. At the same time, according to [73,81,82], many polycations adsorbed on the bacterial surface are able to penetrate into the phospholipid bilayer (membrane lysis mechanism). As a result, the membrane is destroyed and the cell (microorganism) dies.

Therefore, it can be stated that DADMAC–DMAPMA copolymers obtained from a mixture with a molar composition of [DMAPMA]:[DADMAC] = (80÷20):(20÷80) have antibacterial activity against SRB at a concentration of at least 0.05 wt%.

## 4. Conclusions

In this research, DADMAC–DMAPMA copolymers with different molar compositions were synthesized by free radical copolymerization at 343 K in an aqueous solution in the presence of an initiator: (NH_4_)_2_S_2_O_8_. The composition and structure of the synthesized copolymers were determined by FTIR, ^1^H NMR spectroscopy, elemental analysis, and conductometric titration. It was found that the DMAPMA monomer is more active than the DADMAC monomer in the radical copolymerization reaction; thus, the DADMAC–DMAPMA copolymers were always enriched in DMAPMA monomer.

The study of the static shear stress of the bentonite suspension in the presence of the DADMAC–DMAPMA copolymer with different molar compositions showed that this copolymer has structure-formation properties; therefore, it can be used to control the viscosity (rheological properties) of clay drilling fluids.

It was also found that the DADMAC–DMAPMA copolymer with a wide molar composition had an antibacterial activity at a concentration of not less than 0.05 wt% and can be used to inhibit the growth of the corrosion-dangerous bacteria, SRB.

## Data Availability

Not applicable.

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
