# Peer review of "A Novel Cationic Polymer Surfactant for Regulation of the Rheological and Biocidal Properties of the Water-Based Drilling Muds"

_polymers, 2023, doi:10.3390/polym15020330_

Round 1

Reviewer 1 Report

The paper is well organized. The copolymerization of these monomers is interesting and this paper brings information and novelty.

Please highlight the novelty of paper at the end of introduction

Please adapt axes of figures (not necessary full axis).

The conclusion is uncomplete.

Author Response

The Answers to the questions of the Reviewer #1

Dear Reviewer! Thanks a lot for your valuable comments!

  1. The paper is well organized. The copolymerization of these monomers is interesting and this paper brings information and novelty.

  1. Please highlight the novelty of paper at the end of introduction.

Answer to the Question: The following sentence is added at the end of the introduction: “In this work, a new polymeric surfactant has been synthesized, which has a structure-forming and biocidal effect”.

  1. Please adapt axes of figures (not necessary full axis).

Answer to the Question: Some figure axes have been adapted

  1. The conclusion is incomplete.

Answer to the Question:  The conclusion is completed as follows “It was also found that DADMA-DMAPMA copolymer with wide molar composition had an antibacterial activity at the concentration not less than 0.05 wt % and can be used for the inhibition of the growth of corrosion-dangerous bacteria – SRB”.

On behalf of the authors Abdiyev K.Zh.

December 07, 2022

December 07, 2022

Reviewer 2 Report

This paper describes the research on the synthesis and some properties of a new copolymer for the use as surfactant for drilling fluids. The novelty is the using of new comonomer in the free radical copolymerization with N,N-diallyl-N,N-dimethylammonium chloride, i.e. N-[3-(dimethylamino)propyl]-metacrylamide in place of earlier used dimethylacrylamide (Polym. Adv. Technol. 2021 32 2669 and Polymer Sci. Ser. B 2015 57 217).  Some properties of this new copolymer related to its use in drilling muds have been studied. It was shown that the copolymer may control some rheological properties of the muds and give the muds antibacterial properties thus preventing microbiological corrosion of drilling equipment. Results of this research could be interesting for the community of specialists in underground drilling and for some specialists in polymer surfactants. This paper could be considered for publication in Polymers after deep revision regarding my comments placed below.

1.       In Experimental Section (subsection 2.2.4) the description of kinetic studies is not clear. How was the conversion-time dependence obtained? Was it sampling or each point was determined in separate synthetic experiment or a mother solution was prepared. What does it mean “…in an aqueous environment…”. I understand that the copolymerization was performed in a concentrated water solution. Water had to be added to the reaction mixture taking water in DADMAC in consideration. I hope that it was distilled or degassed water.

2.       This research is mostly devoted to the copolymer synthesis. The most important characteristics of copolymerization is the comonomer reactivity ratio which should be determined. The authors should perform the Fineman Ross or Mayo Lewis analysis of results obtained at various comonomer ratios to calculate rDADMAC and rDMAPMA.

3.       There is no information on molecular weight of the new copolymers. I understand that studies by SEC of this type of copolymers may be difficult, but measuring of intrinsic viscosity of the copolymers in water or an alcohol would give an idea of average size of macromolecules and provide information about interaction between macromolecules. The comparison of the viscosity behavior for the copolymers obtained at various monomer and catalyst concentrations would be very interesting.

4.       Figures 5 and 6 show that the copolymerization initially takes fast course and after about 40 minutes strongly slows down for all concentrations of initiator and monomers. The authors should explain what the reason is for this behavior.

5.       The method of calculation of Rc is not clear. It requires the knowledge of dy/dt which is changed dramatically as the reaction proceeds. It is not known at which time of the reaction the slope is measured. It cannot be the initial rate, because the conversion-time has not been studied in the initial period of the reaction (only one experimental point taken at above 80% of the monomer conversion range measured).

6.       The description of 1H NMR spectra requires more explanation because some signals which appear in the spectra of homopolymers do not appear in the spectrum of the copolymer. For example the signal at 1.72 ppm in the spectrum of  poly-DMAPMA attributed to protons at carbon 5 is not seen in the copolymer spectrum although the respective local structure is not changed as showed in Figures 2 and 4. What are signals at 2.95 and 3.06 ppm in the spectrum of poly-DADMAC.

7.       The authors abuse the name rate constant. Symbol k named rate constant is used for elementary reactions, while for composite reactions k should be named specific rate (line 316 and in other places).

8.       In Figure 6 the quantity in ordinate should be marked.

9.       The symbol of time should be unified: t or ꚍ.

10.   Line 476. The sentence is not completed.    

Author Response

The Answers to the questions of the Reviewer # 2

Dear Reviewer! Thanks a lot for your valuable comments!

Reviewer report

Начало формы

This paper describes the research on the synthesis and some properties of a new copolymer for the use as surfactant for drilling fluids. The novelty is the using of new comonomer in the free radical copolymerization with N,N-diallyl-N,N-dimethyl-ammonium chloride, i.e. N-[3-(dimethylamino)propyl]metacrylamide in place of earlier used dimethylacrylamide (Polym. Adv. Technol. 2021 32 2669 and Polymer Sci. Ser. B 2015 57 217).  Some properties of this new copolymer related to its use in drilling muds have been studied. It was shown that the copolymer may control some rheological properties of the muds and give the muds antibacterial properties thus preventing microbiological corrosion of drilling equipment. Results of this research could be interesting for the community of specialists in underground drilling and for some specialists in polymer surfactants. This paper could be considered for publication in Polymers after deep revision regarding my comments placed below.

  1. In Experimental Section (subsection 2.2.4) the description of kinetic studies is not clear. How was the conversion-time dependence obtained? Was it sampling or each point was determined in separate synthetic experiment or a mother solution was prepared. What does it mean “…in an aqueous environment…”. I understand that the copolymerization was performed in a concentrated water solution. Water had to be added to the reaction mixture taking water in DADMAC in consideration. I hope that it was distilled or degassed water.

Answer to the Question # 1:

As mentioned in the experimental part (2.2.4. Synthesis), the copolymerization reactions were carried out in ampoules in an aqueous medium and distilled water was used. Therefore, the conversion of the copolymerization reaction was determined by the mass of the resulting copolymer after a certain time (after 30, 60, .... minutes) for each individual synthetic run. The sentence “…in an aqueous environment” means that the copolymerization reactions are carried out in an aqueous medium (in aqueous solutions of monomers). As noted in section 2.2.4. Synthesis, before each experiment, an aqueous solution of monomer mixtures was purged with an inert gas (argon) for 20 minutes to remove oxygen from the reaction medium.

  1. This research is mostly devoted to the copolymer synthesis. The most important characteristics of copolymerization is the comonomer reactivity ratio which should be determined. The authors should perform the Fineman Ross or Mayo Lewis analysis of results obtained at various comonomer ratios to calculate rDADMACand rDMAPMA.

Answer to the Question # 2:

Yes, we agree with your statement. The study of the kinetics of the copolymerization reaction for this system is the subject of our further work. In the volume of one article it is difficult to cover all aspects of the study. 

  1. There is no information on molecular weight of the new copolymers. I understand that studies by SEC of this type of copolymers may be difficult, but measuring of intrinsic viscosity of the copolymers in water or an alcohol would give an idea of average size of macromolecules and provide information about interaction between macromolecules. The comparison of the viscosity behavior for the copolymers obtained at various monomer and catalyst concentrations would be very interesting.

Answer to the Question # 3:

Yes, of course it is possible to determine the molecular weight of synthesized copolymers from viscometric titration data using the equation [η] = K⋅Ma. However, for a more accurate determination of the molecular weight, we must know the values of the constants K and a for this copolymer system in an aqueous medium or in a saline solution. These constants are not known for the new copolymer. Therefore, we did not engage in the determination of molecular weight by this method. In the future, we plan to determine the molecular weight by mass spectroscopy. 

  1. Figures 5 and 6 show that the copolymerization initially takes fast course and after about 40 minutes strongly slows down for all concentrations of initiator and monomers. The authors should explain what the reason is for this behavior.

Answer to the Question # 4:

The faster increase in conversion in the initial period and the slower increase in the conversion of the copolymerization reaction after 40 minutes can be explained as follows: In the initial period, the monomer concentration is relatively large, and accordingly, the probability of monomer collision with the initiator and macroion is relatively high. Over time, the concentration of monomers in the medium decreases and, accordingly, the probability of collision of monomers with each other and with the initiator will decrease. According to the data in Fig. 5 and 6, the conversion also depends on the concentration of the initiator: with an increase in the concentration of the initiator, the conversion increases.

At a constant concentration of the initiator (old Fig. 6, corrected Fig. 7), the degree of conversion increases with increasing concentration of monomers, for the reason that at a high concentration the probability of collisions of monomers with each other and with macroions increases. For the studied system, the optimal concentration of monomers is 4 mol/L.

  1. The method of calculation of Rc is not clear. It requires the knowledge of dy/dt which is changed dramatically as the reaction proceeds. It is not known at which time of the reaction the slope is measured. It cannot be the initial rate, because the conversion-time has not been studied in the initial period of the reaction (only one experimental point taken at above 80% of the monomer conversion range measured).

Answer to the Question # 5: Rc was calculated according equation (1) [49]:

                                                                                                 (1)

where [M] is the total monomers’ concentration, mol/L; t is the time, s; y is the conversion, %.

The conversion rate (dy/dt ) was calculated from the slope of the dependence y(%) on time (t) (Fig. 5 and 7) in the initial time period (in the time interval 0 - 40 minutes). And the maximum conversion y(max, %) was determined from Fig. 5 and 7 after 240 minutes. 

  1. The description of 1H NMR spectra requires more explanation because some signals which appear in the spectra of homopolymers do not appear in the spectrum of the copolymer. For example, the signal at 1.72 ppm in the spectrum of  poly-DMAPMA attributed to protons at carbon 5 is not seen in the copolymer spectrum although the respective local structure is not changed as showed in Figures 2 and 4. What are signals at 2.95 and 3.06 ppm in the spectrum of poly-DADMAC.

Answer to the Question # 6: The protons of the carbon atom 5 of the poly-DMAPMA homopolymer include not only the peak at 1.72, but also the peaks at 1.51-1.53 (Figure 2 – 1H NMR spectrum of poly-DMAPMA). A peak at 1.53 is also visible in the spectrum of the copolymer (Figure 4 - 1H NMR spectrum of copolymer DADMAC-DMAPMA). In all likelihood, in the copolymer, the intensities of some peaks characteristic of an individual monomer weaken.  

  1. 7.       The authors abuse the name rate constant. Symbol k named rate constant is used for elementary reactions, while for composite reactions k should be named specific rate (line 316 and in other places).

Answer to the Question # 7: Sentence  “reaction rate constant”  was changed with “specific rate” (line 316 and in other places).

  1. In Figure 6 the quantity in ordinate should be marked.

Answer to the Question # 8: the quantity in ordinate is markedю

  1. The symbol of time should be unified: t or ꚍ.

Answer to the Question # 9: The time in the article is marked as tю

  1. Line 476. The sentence is not completed.  

Answer to the Question # 10: The sentence is completed as “It was also found that DADMA-DMAPMA copolymer with wide molar composition had an antibacterial activity at the concentration not less than 0.05 wt % and can be used for the inhibition of the growth of corrosion-dangerous bacteria – SRB”.  

On behalf of the authors Abdiyev K.Zh.

December 07, 2022

Reviewer 3 Report

Manuscript K. Zh. Abdiyev et al. dedicated to researching the rheological and biocidal properties of the water-based drilling muds. In their literary review, the authors at a decent level introduce the reader to the essence of the issue.
And although it is a little more than one page, it contains 47 sources. The result of the theoretical part is the task of obtaining of new water-soluble polymer surfactants. The emphasis is on the use of industrial monomers.
In the course of getting acquainted with the work, I had the main question of how the authors can extrapolate their results to real objects. In my opinion, water should be used from a real source. Accordingly, specific characteristics must be given for it.
It would not hurt to add to the manuscript the data of morphological studies, as well as rheological time and temperature dependences of viscosity and modules.
The authors did not provide a list of keywords
Figure 11. In my opinion, these photographs can be removed from the manuscript as it is very difficult to draw any conclusions from them.
Conclusions. The paragraph breaks - "It was also found that DADMA-DMAPMA copolymer with wide molar compo"

Author Response

The Answers to the questions of the Reviewer # 3

Dear Reviewer! Thanks a lot for your valuable comments!

Reviewer report

Manuscript K. Zh. Abdiyev et al. dedicated to researching the rheological and biocidal properties of the water-based drilling muds. In their literary review, the authors at a decent level introduce the reader to the essence of the issue.
And although it is a little more than one page, it contains 47 sources. The result of the theoretical part is the task of obtaining of new water-soluble polymer surfactants. The emphasis is on the use of industrial monomers.
In the course of getting acquainted with the work,

  1. I had the main question of how the authors can extrapolate their results to real objects. In my opinion, water should be used from a real source. Accordingly, specific characteristics must be given for it.

Answer to the Question: Of course, you are right. Electrolytes can have a significant effect on the rheological properties of water-soluble polymers (polyelectrolytes). Therefore, in the next step, we will  investigate the influence of the nature and concentration of the electrolytes (also sea water).

  1. It would not hurt to add to the manuscript the data of morphological studies,

as well as rheological time and temperature dependences of viscosity and modules.

Answer to the Question:

Static shear stress (SSS, gel strength) is one of the most important mud parameters in drilling. It indicates the rate of formation and strength of the gel that forms in suspension at rest. The higher the SSS, the larger the size of the particles of suspension which will not settle on the bottom of the well. SSS is measured at low shear rate after a period of time. According to the standard API procedure, this is 1 minute and 10 minutes. In this work, we measured SSS after one and ten minutes.

Of course, temperature can have a significant effect on the rheology of a suspension. In the future, we plan to study the effect of temperature on suspension rheology.

  1. The authors did not provide a list of keywords.

Answer to the Question: Keywords were added.

Keywords: copolymer, structure-formation, biocidal polymers, antibacterial activity, sulfate-reducing bacteria

  1. Figure 11. In my opinion, these photographs can be removed from the manuscript as it is very difficult to draw any conclusions from them.

Answer to the Question: Figure 11 was removed from manuscript.

  1. Conclusions. The paragraph breaks - "It was also found that DADMA-DMAPMA copolymer with wide molar compo".

Answer to the Question:  The conclusion is completed as follows “It was also found that DADMA-DMAPMA copolymer with wide molar composition had an antibacterial activity at the concentration not less than 0.05 wt % and can be used for the inhibition of the growth of corrosion-dangerous bacteria – SRB”.

On behalf of the authors Abdiyev K.Zh.

December 07, 2022

Round 2

Reviewer 2 Report

I am not satisfied with the authors’ answers to my remarks. Some of them are enigmatic and some do not provide the significant answer to the posed question.

About 70% of the research described in this paper concerns new copolymerization. The reactivity ratios of co-monomers is the basic knowledge which should be provided. The authors could do this because they studied the copolymerization at different initial proportions of co-monomers which is needed for the determination of these ratios . I do not understand why they have not done the additional step to get these ratios.

I understand that kinetics of copolymerization is complex because the proportion of co-monomers is changed with conversion, but it is very unusual what is seen in Figures 5 and 6. Dramatic change in the reaction rate appears always at the same time of the reaction (about 30 min) independently of the monomer and initiator concentration and independently of the monomer conversion. The trivial answer of authors do not explain this strange behavior.

The authors ignore my question about the determination of derivative dy/dt. From their answer I conclude that the derivative was calculated only from one experimental point as y/t for this point. This is not derivative and gives only its approximation. Rc is approximate rate of the reaction in its initial period. The authors may do this, but it should be clearly explained in the text of paper.

Author Response

The Answers to the questions of the Reviewer # 2
Dear Reviewer! Thank you very much for your valuable comments!

Please find the answer to your comments in the attachment. 

Reviewer 3 Report

The authors answered the questions posed and took into account the comments of the reviewer. Further work can be considered by the editor.

You can add rheology, etc. to the list of keywords.

Author Response

The Answers to the questions of the Reviewer # 3
Dear Reviewer! Thank you very much for your valuable comments!

1. Answer to the suggestion “ You can add rheology, etc. to the list of keywords".

The word “Rheology” was added to the list of keywords

On behalf of the authors Abdiyev K.Zh.

December 23, 2022

Round 3

Reviewer 2 Report

Reactivity ratios are determined which is important improvement of the paper. The overall kinetics is strange, but the major results are based on the initial rate.